# Down-Regulation of Neogenin Decreases Proliferation and Differentiation of Spermatogonia during the Early Phase of Spermatogenesis

**DOI:** 10.3390/ijms232314761

**Published:** 2022-11-25

**Authors:** Jin Woo Park, Yu Jin Kim, Sang Jin Lee, Jung Jae Ko, Dae Keun Kim, Jae Ho Lee

**Affiliations:** 1CHA Fertility Center Seoul Station, Seoul 04634, Republic of Korea; 2Department of Biomedical Sciences, College of Life Science, CHA University, Pochen 11160, Republic of Korea; 3Institute of Animal Genetic Resources Affiliated with Traditional Hanwoo Co., Ltd., Boryeong 33402, Republic of Korea; 4Department of Urology, CHA Fertility Center Seoul Station, CHA University School of Medicine, Seoul 04637, Republic of Korea

**Keywords:** spermatogenesis, neogenin, male infertility, conditional knock-out, CRISPR/Cas9

## Abstract

Non-obstructive azoospermia is a major clinical issue associated with male infertility that remains to be addressed. Although neogenin is reportedly abundantly expressed in the testis, its role in mammalian spermatogenesis is unknown. We systematically investigated the role of neogenin during spermatogenesis by performing loss-of-function studies. Testis-specific neogenin conditional knock-out (cKO) mice were generated using CRISPR/Cas9 and neogenin-targeting guide RNAs. We analyzed the expression profiles of germ cell factors by RT-PCR and Western blotting. Neogenin localized mainly to spermatogonia in seminiferous tubules of mouse testes. RT-PCR and Western blot analyses further demonstrated that neogenin expression varied during spermatogenesis and was dramatically increased at postnatal day 12–25 during the pubertal stage. In neogenin-cKO mouse testes, the ratio of primary and secondary spermatocytes was significantly decreased compared with the control, while the number of apoptotic testicular cells was significantly increased. Taken together, these results suggest that neogenin plays a pivotal role in the maintenance and proliferation of spermatogonia during the early stage of spermatogenesis in mice.

## 1. Introduction

Non-obstructive azoospermia (NOA) is one of the severest forms of male infertility and results from the failure of spermatogenesis [1,2,3,4,5]. Unfortunately, there is no effective treatment for NOA except invasive sperm retrieval [5]. Spermatogenesis is a complex but highly ordered process during which male germ cells proceed through a series of differentiation steps to produce haploid flagellated spermatozoa [3]. This process is dependent on a pool of undifferentiated sperm germ cells called spermatogonia, which commences development of the spermatogenic lineage by making the decision to become progenitor spermatogonia [6]. Subsequently, progenitors acquire the differentiating spermatogonial stem cell phenotype and undergo a series of amplifying mitoses while becoming competent to enter meiosis. These cells then undergo a remarkable transformation from round spermatids to flagellated spermatozoa [7]. Therefore, differentiation of spermatogonial stem cells is an essential step in spermatogenesis [8]. However, many questions persist regarding the regulation of spermatogonia proliferation and differentiation, and the underlying mechanism [4,8,9].

Neogenin, a 190 kDa cell surface receptor protein belonging to the immunoglobulin (Ig) superfamily with four Ig domains and six fibronectin-type III (FnIII) domains that are repeated in its extracellular region [10], was first reported to be densely expressed in the testis in 1999 [11]. However, the role of neogenin in spermatogenesis has not been fully elucidated. Although RT-PCR data indicate that neogenin is highly expressed in the testis, ovary, and uterus [11], it is involved in cell behavioral regulation including neuronal axon growth, morphogenesis, and movement [10,11,12,13,14,15,16,17,18,19,20].

The present study investigated the role of neogenin during mammalian spermatogenesis by attempting to identify which testicular cells most highly express neogenin and exploring its expression profiles during the proliferation and maturation of spermatogonial stem cells. Furthermore, we successfully generated testis-specific neogenin conditional knock-out (cKO) mice using CRISPR/Cas9 to verify the role of neogenin during spermatogenesis in vivo.

## 2. Results

### 2.1. Neogenin Expression Varies during Spermatogenesis

To investigate if neogenin is involved in spermatogenesis, mouse seminiferous tubules were histologically examined. Figure 1A shows hematoxylin and eosin staining of cross-sectioned seminiferous tubules at postnatal day 6–24. At postnatal day 6, seminiferous tubules contained only germ cells with no maturing sperm cells. At postnatal day 12, the only spermatogonia was observed in most seminiferous tubules. At postnatal day 18, spermatogonia start becoming primary spermatocytes and secondary spermatocytes at the end of meiosis, but no mature sperm were evident. At postnatal day 24 or later, fully mature sperm were observed in most seminiferous tubules. Based on these observations, spermatogenesis in mouse testes can be divided into the following three distinct stages: the spermatogonial stage, which spans postnatal day 6–12; the spermatocyte stage, which spans postnatal day 12–18; and the spermiogenesis stage, which spans postnatal day 18–24. RT-PCR (Figure 1B) and Western blot (Figure 1C) analyses revealed that neogenin expression varied during postnatal spermatogenesis, from low to high. Neogenin began to appear from postnatal day 6 and persisted until postnatal day 24. However, neogenin protein expression significantly increased during the primary and secondary spermatocyte stages, and peaked at postnatal day 18–24, which corresponds to the spermiogenesis stage. Neogenin mRNA expression exhibited a comparable pattern and peaked at postnatal day 24. Furthermore, the expression patterns of the germ cell proliferation signaling factors Oct3/4, Sox2, Nanog, Ki67, and TP2 were similar to that of neogenin. In particular, Sox2 was abundantly expressed at the spermatocyte stage, Nanog was highly expressed after the secondary spermatocyte stage, and Oct3/4 expression coincided with neogenin expression during the entire spermatogenesis process. These observations led us to hypothesize that neogenin is involved in the regulation of germ cell proliferation. To verify this, the localization of neogenin in seminiferous tubules was carefully examined by immunohistochemistry. Male germ cells were positive for neogenin using DAB(3,3′-diaminobenzidine) staining, indicating that neogenin is mainly expressed in spermatogonial stem cells during the early phase of spermatogenesis (Figure 1E).

### 2.2. Neogenin Down-Regulation Hampers Differentiation of Primary Spermatocytes into Secondary Spermatocytes

We generated testis-specific neogenin-cKO mice using CRISPR/Cas9 and neogenin-targeting synthetic guide RNA (sgRNA) by direct in vivo transfection into the left testis, while a sham control without neogenin-targeting sgRNA was transfected into the right testis (Figure 2A). The size and morphology of the control testis were normal, while the neogenin-cKO testis was approximately 30–40% smaller (Figure 2B). At 14 days after in vivo transfection, the neogenin-cKO testis weighed significantly less than the control testis (0.14 ± 0.03 g vs. 0.12 ± 0.02 g, n = 20) (Figure 2C). Histochemical examination of seminiferous tubules by hematoxylin and eosin staining revealed that the neogenin-cKO testis contained markedly fewer testicular cells (Figure 2D), indicating that down-regulation of neogenin arrests testicular cell proliferation in seminiferous tubules.

cKO of neogenin dramatically decreased the numbers of primary and secondary spermatocytes in seminiferous tubules (Figure 2D). In addition, the immunohistochemical data shown in Figure 1E and Figure 2E indicated that neogenin specifically localized to spermatogonial germ cells, and cKO of neogenin hampered early meiotic processing of primary spermatocytes into secondary spermatocytes. This clearly implies that neogenin is involved in the early phase of the spermatogenesis differentiation process.

### 2.3. Expression Profiles of Spermatogonial Stem Cell Markers in Neogenin-cKO Testes

We investigated whether neogenin is involved in the maintenance and proliferation of spermatogonia in seminiferous tubules. Real-time qPCR (Figure 3A) and Western blot (Figure 3B) analyses were performed to verify that the level of neogenin is closely correlated with those of Oct3/4, Sox2, Nanog, Ki67, and TP2, which are markers of undifferentiated and pluripotent cells. As expected, upon down-regulation of neogenin in the testis, Oct3/4, Sox2, and Nanog were also down-regulated. Expression of Ki67, a proliferation marker, and TP2, a mature sperm marker, was also lower in the neogenin-cKO testis than in the control testis.

### 2.4. Apoptosis in Neogenin-cKO Testes

We investigated whether apoptosis is responsible for the decrease in the number of spermatogonia upon down-regulation of neogenin. The TUNEL assay demonstrated that the number of apoptotic cells in seminiferous tubules was greatly increased upon down-regulation of neogenin compared with the control (Figure 4A). Quantitatively, the number of apoptotic cells was increased by more than 90% when neogenin was down-regulated (Figure 4B).

## 3. Discussion

This study demonstrated that neogenin plays a pivotal role during the early phase of spermatogenesis. Neogenin specifically localized to spermatogonia at the base of seminiferous tubules and its expression pattern varied depending on the phase of postnatal development including the early spermatogonial phase (postnatal day 12 or less) and primary and secondary spermatocyte phases (postnatal day 18–24). Additionally, immunohistochemistry and Western blot analyses using an anti-neogenin antibody showed that neogenin was abundantly expressed during the pubertal phase at postnatal day 18–24 to initiate proliferation and differentiation of spermatogonia. Spermatogenesis has three phases: proliferation of spermatogonia only phase, first meiosis as primary spermatocyte and second meiosis as secondary spermatocyte phase, final phase as spermiogenesis for haploid maturation to sperm [8]. Our data reveal expression of neogenin increase at the postnatal day 12 to 24 for proliferation and meiosis stage for differentiation of spermatocytes.

RT-PCR and Western blot analyses revealed that expression of neogenin was similar to that of Oct3/4, Sox2, and Nanog, which are pluripotency markers in spermatogonia [6,9,21,22]. Therefore, neogenin may influence spermatogonia behavior via stem cell regulatory factors. One plausible possibility is that neogenin functions as a key receptor in a signal transduction pathway that affects expression of Oct3/4, Sox2, and Nanog during spermatogenesis [4]. Accordingly, Oct3/4 is implicated to function in the early phase of spermatogenesis as a key factor for spermatogonia [4,7,9,21]. We previously reported that neogenin is involved in regulation of Oct3/4, Sox2, and Nanog, which are signaling molecules in pluripotent cells in early embryos [23].

More insight into the role of neogenin in spermatogenesis was provided by assessment of functional and morphological differences between control and neogenin-cKO testes. We used CRISPR/Cas9 to disturb neogenin expression specifically in the testis. This was crucial because neogenin plays multiple roles, such as those in proliferation, differentiation, and apoptosis, depending on the tissue and cell type. Additionally, neogenin has several ligands that are involved in various types of cell-specific regulation. The neogenin-cKO testis was smaller and weighed less than the control testis. Thus, many spermatogonia seemed to be lost from seminiferous tubules in the neogenin-cKO testis. These loss-of-function studies revealed a clear role of neogenin in seminiferous tubules that is attributable to regulation of expression of stem cell factors such as Oct3/4, Sox2, and Nanog during the early spermatogonial phase. Consistently, stem cell factor expression was decreased in the neogenin-cKO testis. Spermatogenesis is thought to be preceded by maintenance of spermatogonia though their proliferation. Oct3/4, Sox2, and Nanog regulate this proliferation by functioning as transcription factors. Several other factors that are important for the proliferation and maintenance of undifferentiated spermatogonia, such as Ki67 and TP2, have been described [24,25], in addition to Oct3/4, Sox2, Nanog, and LIN28, which also regulate pluripotency in mammalian stem cells. Some of these factors are expressed in both germ and somatic cells in the testis [26]. Oct3/4 have expressed within the spermatogonia and thereby regulate the epigenetic state of undifferentiated cells [27]. neogenin also has been found in the developing testis [11]. but the role of neogenin was not clearly identified in spermatogenesis.

Testicular cKO of neogenin significantly increased the number of apoptotic cells in seminiferous tubules, and spermatogonia displayed strong apoptotic signals. Neogenin is involved in survival and proliferation of spermatogonia at an early phase of spermatogenesis. Apoptosis in the testis is a normal event when proliferation and differentiation of a germ cell are arrested. Some toxic materials and defects of growth factors arrest spermatogenesis, and arrested germ cells undergo apoptosis in seminiferous tubules [28]. Therefore, cKO of neogenin in spermatogonia may lead to loss of signaling necessary for maintenance of stem cells, causing spermatogenesis to arrest in germ cells. In support of this, mRNA and protein expression of neogenin in the human testis has been documented in GTEx (https://www.gtexportal.org/ home/gene/NEO1) and a proteomics database (https://www.proteinatlas.org/ENSG00000278646-AC008162.2/tissue/testis#img). Therefore, neogenin needs further study whether involved or not in the arrest of human spermatogonia for overcoming male infertility.

## 4. Materials and Methods

### 4.1. Preparation of Mouse Testes

All experimental procedures for animal breeding, care, and sacrifice were performed in compliance with the regulations of the Institutional Animal Care and Use Committee (IACUC No. IACUC170178) of Sahmyook University, Seoul, South Korea. Mouse testes (ICR strain) were surgically removed at postnatal day 6, 12, and 18 and at 24 weeks of age after anesthetization with a mixture of zoletil (30 mg/kg) and rompun (10 mg/kg). Harvested testes were fixed with 4% paraformaldehyde for histological examination or stored at −80 °C until RT-PCR and Western blot analyses were performed.

### 4.2. Histochemical Examination of Testes

Each mouse testis was fixed with 4% paraformaldehyde overnight at 4 °C. Paraffin embedding and microtoming were then performed to obtain 5 µm thick cross-sections of the testis. For histological analysis, hematoxylin and eosin staining was performed following a standard protocol. For immunohistochemical staining, testicular tissue-containing paraffin sections on glass slides were deparaffinized with a xylene series ranging from 100% to 75% for 10 min each, and then rehydrated with an ethanol series ranging from 100% to 75% for 10 min each at room temperature. For antigen retrieval, the tissue sections were boiled in 0.01 M citrate buffer (pH 6.0) for 20 min in an autoclavable jar. After permeabilization with phosphate-buffered saline (PBS) containing 0.1% Triton-X100 for 20 min at room temperature, the sections were washed three times with fresh PBS. Subsequently, the tissue sections were incubated overnight at 4 °C with a rabbit anti-neogenin primary antibody (NB1-89651; Novus Biologicals, Centennial, CO, USA) diluted 1:100 in PBS containing 0.03% bovine serum albumin (BSA). For DAB staining, a horseradish peroxidase-conjugated secondary antibody was applied to tissue sections, and samples were counterstained with hematoxylin for 5 min at room temperature. Slides were mounted using mounting media and covered with coverslips. Tissue slides subjected to DAB staining were imaged using an inverted light microscope (Eclipse Ti2; Nikon, Tokyo, Japan) equipped with a camera (DS-Ri2, Nikon) and imaging software (NIS-Elements ver. 4.4., Nikon).

### 4.3. PCR for Gene Expression Analysis

To extract mRNA, 1.0 mL TRIzol solution (15596026; Invitrogen, Carlsbad, CA, USA) was added to 1.0 g testicular tissue according to the manufacturer’s instructions. Total RNA was reverse-transcribed into cDNA using AccuPower^®^ CycleScript RT PreMix (K-2050; Bioneer, Daejeon, Republic of Korea) and then amplified with AccuPower^®^ Taq PCR PreMix (K-2602, Bioneer) using primer sets specific for mouse neogenin, Oct3/4, Nanog, Sox2, and β-actin (Appendix A). In total, 10 pmol/µL forward and reverse primers and 200 ng template cDNA were added to AccuPower^®^ Taq PCR PreMix tubes, and then distilled water was added up to a total volume of 20 µL. The PCR cycling conditions were as follows: 1 min at 95 °C, followed by 30 cycles of denaturation for 30 s at 95 °C, annealing for 30 s at 60 °C, and extension for 50 s at 72 °C. PCR products were resolved on 1.5% agarose gels with Safeview™ FireRed (G926; Applied Biological Materials, Richmond, BC, Canada). The agarose gel was visualized under ultraviolet illumination using a gel documentation system (WSE-6100 LuminoGraph; ATTO, Tokyo, Japan). Quantitative real-time PCR was performed using AccuPower GreenStar qPCR PreMix (K-6212, Bioneer) and primer sets specific for mouse neogenin, Oct3/4, Nanog, Sox2, and β-actin (Appendix A) on a spectrofluorometric thermal cycler (CFX96 Touch Real-Time PCR Detection System; Bio-Rad, Hercules, CA, USA). The reaction contained 10 pmol/µL forward and reverse primers, 200 ng template cDNA, and AccuPower GreenStar qPCR PreMix. Distilled water was added up to a final volume of 20 µL. The qPCR cycling conditions were as follows: 3 min at 95 °C, followed by 40 cycles of denaturation for 10 s at 95 °C, and annealing and extension for 20 s at 60 °C. Expression of each gene was normalized to that of β-actin. All experiments were repeated three times for statistical analysis (Bio-Rad CFX Maestro 2.3, Bio-Rad).

### 4.4. Western Blotting for Protein Expression Analysis

About 1.0 g testicular tissue was homogenized with 1.0 mL protein extraction buffer (17081; iNtRON Biotechnology, Seongnam, Republic of Korea). Homogenized samples were centrifuged at 15,000 rpm for 30 min at 4 °C and supernatants were collected. The resulting protein extracts were boiled with 4× Laemmli buffer (1610747, Bio-Rad), resolved by 8% SDS-PAGE, and transferred to nitrocellulose membranes (BR162-0112, Bio-Rad). The membranes were incubated for 1 h at room temperature in blocking buffer, which comprised Tris-buffered saline (TBS) supplemented with 0.5% Tween 20 and 5% BSA, and then incubated with the following primary antibodies: rabbit anti-neogenin (NBP1-89651, Novus Biologicals), rabbit anti-Oct3/4 (sc-9081; Santa Cruz Biotechnology, Dallas, TX, USA), goat anti-Sox2 (sc-17319, Santa Cruz Biotechnology), mouse anti-Nanog (sc-134218, Santa Cruz Biotechnology), rabbit anti-TP2 (3941; Biovision, Waltham, MA, USA), rabbit anti-Ki67 (ab15580; Abcam, Cambridge, UK), and mouse anti-β-actin (MA5-15739; Invitrogen, Waltham, MA, USA) overnight at 4 °C. After four washes with TBS containing 0.5% Tween 20 for 5 min each, the membranes were incubated with horseradish peroxidase-conjugated goat anti-mouse (62-6520; Invitrogen, Waltham, MA, USA), goat anti-rabbit (31460; Thermo Fisher, Waltham, MA, USA), and rabbit anti-goat (A27014, Thermo Fisher) antibodies. Immunoreactive bands were visualized using enhanced chemiluminescence (Miracle-Star™, 16028, iNtRON Biotechnology). Bands were imaged using a gel documentation system (WSE-6100 LuminoGraph I, ATTO).

### 4.5. cKO of Neogenin in the Testis

To conditionally knock-out neogenin in the testes of mice (ICR strain) (n = 20), neogenin-targeting sgRNA (Gene code: NM_001042752) was designed using the webtool CRISPR-ERA (http://crispr-era.stanford.edu accessed on 24 March 2018) (Appendix A) following the manufacturer’s protocols. A mixture of CRISPR/Cas9 gene constructs and neogenin-targeting sgRNA was prepared by dissolving 50 μg of each DNA in 400 μL of sterile 5% glucose solution containing trypan blue dye (0.1%) as an indicator. The resulting master mix was gently vortexed and centrifuged briefly. About 6 μL of TurboFect in vivo transfection reagent (R0533, Thermo Fisher) was added to 4.0 μL of the diluted DNA master mix. The mixture was immediately subjected to brief pipetting and incubated for 15–20 min at room temperature. This mixture was microinjected into the left testis of a mouse at postnatal day 12 using a sharp pulled microcapillary glass needle (2 µm ID, 4 µm OD) on a micromanipulator-driven injector (IM-6; Narishege, Tokyo, Japan). As a sham control, the right testis received an equal amount of transfection reagent and CRISPR-Cas9 plasmids but no sgRNA. Twenty-five mice underwent microinjection and 20 mice survived. Both testes were surgically excised to analyze spermatogenesis at 2 weeks after the injection, which was equivalent to postnatal day 26. The phenotypes of testicular cells from neogenin-cKO mice were analyzed by several methods including histological and mRNA and protein expression analyses.

### 4.6. Analysis of Apoptosis

Ten sections of each testis from five mice were evaluated. Apoptosis of testicular cells was measured using an in situ MEBSTAIN^®^ Apoptosis TUNEL Kit (8445; MBL International, Woburn, MA, USA). Paraffin-embedded testicular tissue sections were dewaxed and rehydrated as described in the preceding section and treated with 20 µg/mL proteinase K prepared in 10 mM Tris/HCl (EO0491; Life Technologies, Waltham, MA, USA) for 20 min at room temperature. The tissue sections were then permeabilized in 0.1% Triton X-100 and 0.1% sodium citrate on ice for 2 min, and rinsed twice with PBS. The TUNEL staining reaction was performed according to the manufacturer’s instructions. Some tissue sections were treated with 1 µg/mL DNase I for 10 min at room temperature to introduce DNA strand breaks, which served as positive controls, whereas other tissue sections were treated with the TUNEL mix without terminal deoxynucleotidyl transferase, which served as negative controls. Fluorescein-dUTP was used to label DNA strand breaks. After staining with fluorescein, the tissue sections were counterstained with 0.5 μg/mL propidium iodide (PI) for 5 min, mounted with anti-fade mounting medium (VECTASHIELD^®^ H-1000; Vector Laboratories, Burlingame, CA, USA) under coverslips, and imaged using a confocal microscope (LSM880, Carl Zeiss AG, Oberkochen, Germany).

### 4.7. Statistical Analysis

All experiments were repeated more than three times. Statistical analysis was performed using Sigma plot 12.5 software. The Student’s *t*-test was used to determine statistical significance. The significance level was set at * *p* < 0.05 and *** *p* < 0.001.

## 5. Conclusions

Neogenin specifically localizes to spermatogonia and is abundantly expressed during the pubertal phase spanning postnatal day 18–24. cKO of neogenin arrests spermatogenesis due to failure of spermatogonia to proliferate and differentiate. Consistently, neogenin regulates male germ cell proliferation signaling factors such as Oct3/4, Sox2, Nanog, and Ki67. This indicates that neogenin is required for maintenance and proliferation of spermatogonia. CRISPR/Cas9 can be utilized to conditionally knock-out expression of a gene specifically in testicular tissue. This in vivo direct gene editing tool has great potential to investigate signaling factors involved in spermatogenesis.

## Figures and Tables

**Figure 1 ijms-23-14761-f001:**
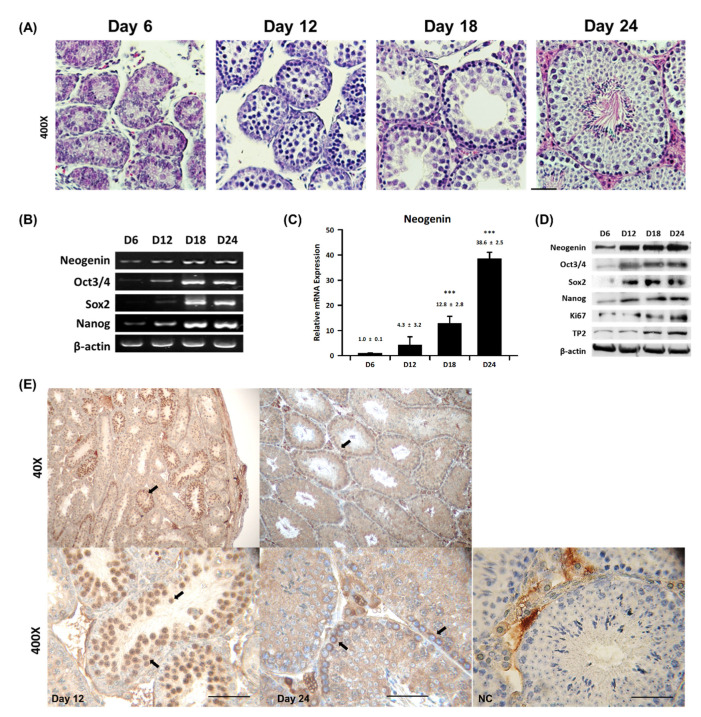
Expression profile of neogenin in the mouse testis. (**A**) Testes obtained from mice at postnatal day 6–24 were cross-sectioned, stained with hematoxylin and eosin, and viewed under a microscope. Scale bar = 100 mm. (**B**) mRNA was extracted from testicular tissues, subjected to RT-PCR to amplify mRNAs encoding neogenin, Oct3/4, Sox2, and Nanog, and resolved on an agarose gel. (**C**) qPCR data of neogenin depend on the postnatal days. (n = 3) (**D**) Proteins were extracted from testicular tissues and analyzed by Western blotting using antibodies against neogenin, Oct3/4, Sox2, Nanog, Ki67, and TP2. β-actin was used as a control. (**E**) Seminiferous tubules were cross-sectioned and subjected to immunohistochemical DAB staining for neogenin at postnatal days 12 and 24. NC, negative control. Arrows indicate neogenin-positive cells. Scale bar = 100 mm. *** *p* < 0.001.

**Figure 2 ijms-23-14761-f002:**
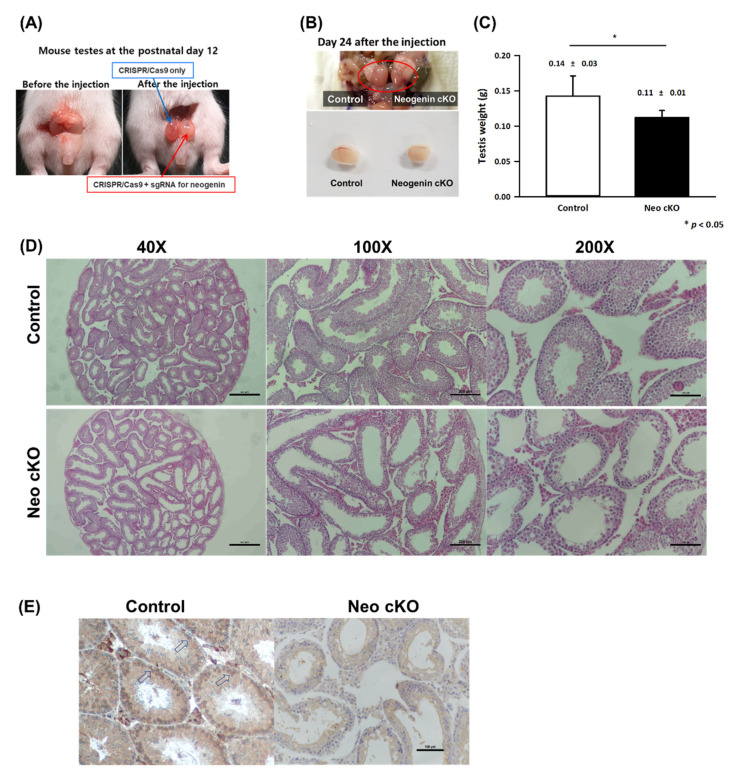
Morphological and cellular comparison of neogenin-cKO and sham control testes. Neogenin was conditionally knocked out in the left testis by CRISPR-Cas9 as described in the Materials and Methods, while the right testis was the sham control. (**A**) External morphology of the testes at postnatal day 12 before and after microinjection. (**B**) External morphology of the testes at 24 days after microinjection. (**C**) Both testes were surgically excised and weighed at 14 days after microinjection (n = 20). * *p* < 0.05 versus control by the Student’s *t*-test. (**D**) Hematoxylin and eosin staining of seminiferous tubules. (**E**) Immunohistochemical DAB staining for neogenin at seminiferous tubules in the testis. Control: control testis; neogenin-cKO: neogenin conditional knock-out testis. Scale bar = 500 μm (40×), 200 μm (100×), and 100 μm (200×).

**Figure 3 ijms-23-14761-f003:**
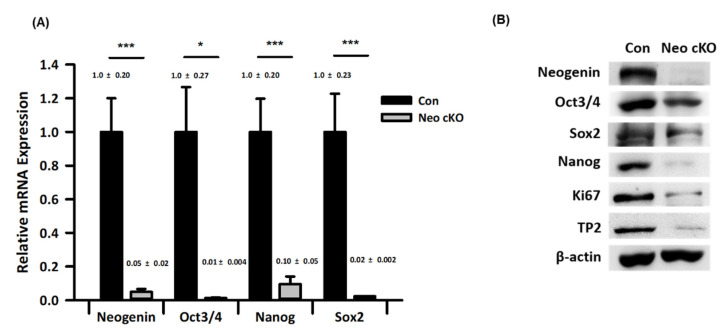
Expression profiling of spermatogonial stem cell markers in control and neogenin-cKO testes. Neogenin was conditionally knocked out in the mouse testis by CRISPR-Cas9 as described in the Materials and Methods. (**A**) Quantitative real-time PCR analysis of relative mRNA expression of neogenin, Oct3/4, Sox2, and Nanog normalized against that of β-actin. (n = 3) (**B**) Proteins extracted from testicular tissues at postnatal day 26 were analyzed by Western blotting. β-actin was used as a housekeeping protein. Con: control testis; Neo cKO: neogenin conditional knock-out testis. Data are the mean ± SD of triplicates. * *p* < 0.05 and *** *p* < 0.001 versus control by the Student’s *t*-test.

**Figure 4 ijms-23-14761-f004:**
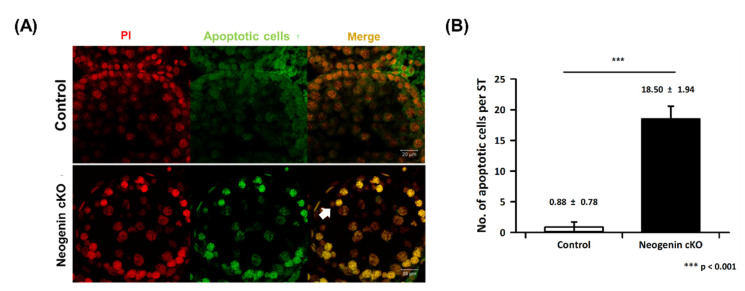
Quantification of apoptotic cell death in seminiferous tubules. Apoptotic death of seminiferous tubular cells was fluorescently measured using the TUNEL assay as described in the Materials and Methods. (**A**) Images of PI-positive nuclei (red) and apoptotic cells (green) were merged. (**B**) The numbers of apoptotic cells (white arrow) were counted and plotted (n = 10). *** *p* < 0.001 versus control by the Student’s *t*-test.

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
