# Peer review of "Down-Regulation of Neogenin Decreases Proliferation and Differentiation of Spermatogonia during the Early Phase of Spermatogenesis"

_ijms, 2022, doi:10.3390/ijms232314761_

Round 1
Reviewer 1 Report
General
The manuscript “Down-regulation of neogenin is decline of the proliferation and differentiation of spermatogonia at early phase of spermatogenesis” by Park and collaborators describes localization of neogenin in postnatal mouse testis and effects of local knock out of Neo1 on testicular development. Neogenin is a context-dependent membrane receptor regulating developmental processes of several tissues and has previously been presented to be abundant in testis. The authors show that in Neo1- knockout testes there are more apoptosis and spermatogenesis has arrested. Etiology of non-obstructive azoospermia often remains unsolved and therefore it is very important to identify candidate factors. Though the original objective of characterizing the role of neogenin in testicular development/spermatogenesis is interesting I have some major concerns about the manuscript.
Major
1. Setup of the experiment: is the aim to study the role of neogenin during the postnatal period until adulthood or during spermatogenesis per se as it could be understood from the text? The authors may have used different postnatal developmental stages to mimic spermatogenesis in adult, but that should be explained and discussed. Do the postnatal stages correspond to stages of spermatogenesis in mature mouse testes that are under complete hormonal regulation? The authors should make difference between development towards a functionally mature testis and spermatogenesis (as it takes place in an adult individual.
2. The execution and follow-up of in vivo knockout. To confirm the specificity of Neo1 knockout the authors should have used two different gRNAs for Neo1 (if possible) leading to similar results, and non-targeting gRNA negative control in a sham setup. They should also have presented genomic DNA sequencing data of knockout testes to prove knockout. I understand that the result could have been mixture of several sequences, but it should have been able to show that indels have taken place in the gene area targeted. Most obviously, the injection is a challenging procedure, since 8 of 20 mice did not survive, but therefore all the proper controls are even more important to rule out effects of inflammation or other unrelated factors. The other methods have been described in detail, but the knockout procedure lacks crucial ones like the sequence of gRNA used.
3. Analysis of mice. What is the number of mice analysed for the Fig.2C? According to the main text 6 testes (3 knockout and 3 control?), but according to the figure text 20 testes have been included in the analysis. N-values should also present according to number of mice not testis, since that shows number of replicates in each group. Instead of columns, it would be more informative to show the data with symbols and lines, where knockout and control testis from a mouse have been linked together. Weight is a very informative parameter for underdevelopment of testis. Analysis of all mice could help to estimate roughly the success of knock out in all of them. Fig. 2 should also be completed with immunohistochemical data with the marker genes to be able to follow the stages of spermatogenesis in knockout testes.
4. Quality of the histological images. There should have been access to original high-resolution images to judge them. In particular, Fig. 1D and 2D should have been shown in more detail to confirm localization of neogenin and the phase of spermatogenetic arrest.
5. The tables were missing from the manuscript. Table 1 and Supplementary Table 2 were mentioned in the main text, but they were not included in the manuscript pdf-file or the other files to download. Also, the files “Original Images for Blots/Gels” was same as “Non-published Material”.
6. The authors could have discussed more about function of neogenin in other tissues and why is had selected to be investigated (in addition that the gene is expressed in testis). In addition, the cellular mechanism of action of neogenin would have added value of the introduction and discussion.
7. Conclusions. The authors claim that “...revealed a clear role for neogenin that is attributable to the regulation stem cell factor expression...” The authors however do not show evidence of causality. The explanation for lack of certain factors could be lack of certain cell types due to the spermatogenesis arrest not a specific effect of neogenin on expression of these gene.
8. Language. Editing by a native English speaker and compacting the text could significantly improve the manuscript. Phrases and words such as “is decline of” (the title), “conjucture” (meaning hypothesis?), “permission” (line 189) in their context may not be clear, for example. There is also inconsistency in use of tenses (such as lines 249 and 320). There are obscure sentences without a verb (such as lines 182-3). There is also repetition (such as describing markers of spermatogonia) and uninformative introductory sentences (line 354, “Finally, a word in needed...”) to give some examples.
9. The references to the databases (lines 221-223) neither add value of the manuscript nor support its hypotheses in the present form.
10. References to published data are unclear, if not misleading. See e.g. lines 206-210. Neogenin has not been mentioned in the references [7], [21] or [27] as it could have been expected from the text.
Minor
1. Line 17, NOA, should be non-obstructive azoospermia (not none).
2. Line 306, sgRNA usually means synthetic guide RNA.
3. References are mainly from the first decade of the millennium. Maybe newer informative references would have been available?
Author Response
General
The manuscript “Down-regulation of neogenin is decline of the proliferation and differentiation of spermatogonia at early phase of spermatogenesis” by Park and collaborators describes localization of neogenin in postnatal mouse testis and effects of local knock out of Neo1 on testicular development. Neogenin is a context-dependent membrane receptor regulating developmental processes of several tissues and has previously been presented to be abundant in testis. The authors show that in Neo1- knockout testes there are more apoptosis and spermatogenesis has arrested. Etiology of non-obstructive azoospermia often remains unsolved and therefore it is very important to identify candidate factors. Though the original objective of characterizing the role of neogenin in testicular development/spermatogenesis is interesting I have some major concerns about the manuscript.
Major
- Setup of the experiment: is the aim to study the role of neogenin during the postnatal period until adulthood or during spermatogenesis per se as it could be understood from the text? The authors may have used different postnatal developmental stages to mimic spermatogenesis in adult, but that should be explained and discussed. Do the postnatal stages correspond to stages of spermatogenesis in mature mouse testes that are under complete hormonal regulation? The authors should make difference between development towards a functionally mature testis and spermatogenesis (as it takes place in an adult individual.
- R) Thank you for your comment. Our study aimed to determine whether neogenin is involved in the initial differentiation of spermatogonia. Postnatal day 6 is when spermatogenesis is at the earliest stage and before the spermatogenesis activation stage. Therefore, we chose postnatal day 6, which is before the pubertal stage, this time is starting spermatogenesis which is help to spermatogenesis mechanism of early stage.
- The execution and follow-up of in vivoknockout. To confirm the specificity of Neo1 knockout the authors should have used two different gRNAs for Neo1 (if possible) leading to similar results, and non-targeting gRNA negative control in a sham setup. They should also have presented genomic DNA sequencing data of knockout testes to prove knockout. I understand that the result could have been mixture of several sequences, but it should have been able to show that indels have taken place in the gene area targeted. Most obviously, the injection is a challenging procedure, since 8 of 20 mice did not survive, but therefore all the proper controls are even more important to rule out effects of inflammation or other unrelated factors. The other methods have been described in detail, but the knockout procedure lacks crucial ones like the sequence of gRNA used.
- R) We have updated sgRNA injection procedure in the materials and methods. The sequence of the neogenin gene is very long; therefore, we performed knock-out using sgRNA targeting multiple sites of Neo1. And we have provided the sgRNA Neo1 information in the supplemental material. Please accept our apologies for previously omitting the supplemental file. We have also updated the Materials and Methods section regarding gene knock-out. Mice at postnatal day 6 are very small body consequently. Some mice have difficult to overcome surgery damage and the dose of anesthesia; therefore, some mice died. We added more experiment.
- 3.Analysis of mice. What is the number of mice analysed for the Fig.2C? According to the main text 6 testes (3 knockout and 3 control?), but according to the figure text 20 testes have been included in the analysis. N-values should also present according to number of mice not testis, since that shows number of replicates in each group. Instead of columns, it would be more informative to show the data with symbols and lines, where knockout and control testis from a mouse have been linked together. Weight is a very informative parameter for underdevelopment of testis. Analysis of all mice could help to estimate roughly the success of knock out in all of them. Fig. 2 should also be completed with immunohistochemical data with the marker genes to be able to follow the stages of spermatogenesis in knockout testes.
- R) We used 12 mice for histological and gene and protein expression analyses. We attempted immunohistochemical staining, but there was a non-specific signal in the testis. We have updated the figure 1E and 2E.
- Quality of the histological images. There should have been access to original high-resolution images to judge them. In particular, Fig. 1D and 2D should have been shown in more detail to confirm localization of neogenin and the phase of spermatogenetic arrest.
- R) We have provided better images to confirm localization of neogenin.
- The tables were missing from the manuscript. Table 1 and Supplementary Table 2 were mentioned in the main text, but they were not included in the manuscript pdf-file or the other files to download. Also, the files “Original Images for Blots/Gels” was same as “Non-published Material”.
- R) We apologize for the missing supplemental file. We have provided a new supplemental file with the revision.
- The authors could have discussed more about function of neogenin in other tissues and why is had selected to be investigated (in addition that the gene is expressed in testis). In addition, the cellular mechanism of action of neogenin would have added value of the introduction and discussion.
- R) Thank you for your comment. We have discussed the role of neogenin in other tissues in the main text.
- Conclusions. The authors claim that “...revealed a clear role for neogenin that is attributable to the regulation stem cell factor expression...” The authors however do not show evidence of causality. The explanation for lack of certain factors could be lack of certain cell types due to the spermatogenesis arrest not a specific effect of neogenin on expression of these gene.
- R) We have provided new images of histochemical staining with an anti-neogenin antibody in Figure 2E.
- Language. Editing by a native English speaker and compacting the text could significantly improve the manuscript. Phrases and words such as “is decline of” (the title), “conjucture” (meaning hypothesis?), “permission” (line 189) in their context may not be clear, for example. There is also inconsistency in use of tenses (such as lines 249 and 320). There are obscure sentences without a verb (such as lines 182-3). There is also repetition (such as describing markers of spermatogonia) and uninformative introductory sentences (line 354, “Finally, a word in needed...”) to give some examples.
- R) The text has been edited by a native English speaker.
- The references to the databases (lines 221-223) neither add value of the manuscript nor support its hypotheses in the present form.
- R) We have checked the references.
- References to published data are unclear, if not misleading. See e.g.lines 206-210. Neogenin has not been mentioned in the references [7], [21] or [27] as it could have been expected from the text.
- R) We check-up the reference for the manuscript.
Minor
- Line 17, NOA, should be non-obstructive azoospermia (not none).
- Line 306, sgRNA usually means synthetic guide RNA.
- References are mainly from the first decade of the millennium. Maybe newer informative references would have been available?
- R) We have revised the text accordingly.
Reviewer 2 Report
The manuscript entitled “Down-regulation of neogenin is decline of the proliferation and differentiation of spermatogonia at early phase of spermatogenesis” brings up the idea of a novel role of neogenin on spermatogonia proliferation and differentiation. Overall, the experiments presented were well performed, although some important points should be revised and taken in consideration by the authors.
Major Comments:
1. Starting by the title, I suggest the author to revise it, since there is no direct cell proliferation analysis presented on the results.
2. Please, inform in the text or in form of supplementary material all the sequences and references (when not design by the authors) for the primers used for RT-PCR, qPCR and CRISPR/Cas9 cKO.
3. Suggested experiment: I strongly suggest performing qPCRs to sustain the affirmation that “level of neogenin expression varied over the postnatal spermatogenesis stages, from low to very high”. Taking the RT-PCR image as base, I disagree that neogenin expression goes from low to VERY high. A qPCR assay would, quantitatively, answer this point.
4. Lines 81-82: primary and secondary spermatocyte stages reaching the highest level between postnatal days 18 and 24, which corresponds to the spermiogenesis stage.
Comment: If that’s the case, I strongly suggest a new figure containing high magnification (1000X) versions of the immunohistochemistry, now including day 18.
5. Suggested experiment: Lines 90-93: To further verify these conjectures, the localization of neogenin expression in the seminiferous tubules were carefully examined by immunohistochemistry. The results shown in Figure 1D revealed that DAB positive stained on a male germ cell, indicating that neogenin is mainly expressed in spermatogonial stem cells at the early phase of the spermatogenesis.
Comment: The immunohistochemistry does not answer the question if neogenin is implicated in germ cell proliferation. For that I suggest an experiment of cell proliferation (BrDU incorporation, for instance) using your cKO against the wild type.
6. Lines 118-119: Shown in Figure 2D is that neogenin-cKO dramatically decreased both the primary and secondary spermatocytes in the seminiferous tubules compared with control groups. In addition, the immunohistochemical data shown in Figure 1D and 2D indicated that neogenin specifically localized at the spermatogonia germ cells, and the downregulation of neogenin by neogenin-cKO deterred early meiotic processing of the primary to the secondary spermatocytes phase, strongly implying that neogenin is involved in the early phase of the spermatogenesis differentiation process.
Suggestion: It is EXTREMELY important to show high magnification figures when affirming that a certain type o germ cell is missing, or it is expressing neogenin. Reformulate figures 1D and 2D.
7. Figure 3: The qPCR results for OCT3/4 and SOX2 are not corroborated by the results of Western Blot.
8. Not all the references are following the same formating model. Please correct that.
Minors:
Lines 37-39: The spermatogenesis is a complex but highly ordered process by which male germ cells proceed through a series of differentiation steps to produce haploid flagellated spermatozoa
Comment: spermatogenesis is a complex AND highly ordered process. Also, prior a series of differentiation there is a mitotic phase, where there is an expansion of germ cells but not differentiation. Please, reformulate this sentence.
Lines 39-40: Underlying this process is a pool of undifferentiated sperm germ cells called spermatogonia
Comment: Not all spermatogonia are undifferentiated. In fact, the identity of undifferentiated spermatogonia is subject of debate. Since the term “spermatogonia” is used to assign single “potentially” undifferentiated spermatogonia to type B spermatogonia, this sentence sounds incorrect.
Lines 42-43: differentiating spermatogonial stem cell phenotype.
Comment: spermatogonia is either differentiating or undifferentiated / stem cell. Therefore, the term “differentiating spermatogonia stem cell might be inaccurate.
Lines 49-52: Neogenin, a 190-kDa cell-surface receptor protein belonging to the immunoglobulin (Ig) superfamily with four Ig domains and six fibronectin-type III (FnIII) domains that are repeated in its extracellular region [10], was first reported to be densely expressed in the testis in 1999 [11]; but until now, the role of neogenin implicated in the spermatogenesis has not been fully elucidated.
Comment: This sentence is long and, because of that, confusing. Please break the sentence into two. Maybe with a stop after “… region [10].
Lines 68-69: At day 6, the seminiferous tubules contained only germ cells with no maturing sperm cells
Comment: I suggest reformulating this sentence, since not only germ cells are present in the seminiferous tubules. We also have Sertoli cells and PTM cells. Therefore, this sentence is essentially incorrect.
Lines 73-76: Based on these observations, the whole spermatogenesis procedure in the mouse testis could be divided into three distinctive stages: the spermatogonial stage which extends from postnatal days 6 to 12; the spermatocyte stage spanning postnatal days 12 to 18; the spermiogenesis stage spanning postnatal days 18 to 24.
Comment: That is a very problematic sentence. It’s known that in all organisms studied so far that, didactically, spermatogenesis is divided in three phases: spermatogonial, meiotic, and spermigenic. Therefore, to conclude from your images that in mice, spermatogenesis is divided in these three phases from day 6-24 is totally irrelevant.
Lines 87-89: Nanog was highly expressed after the secondary spermatocyte stage while Oct3/4 expression coincided with neogenin expression during the entire spermatogenesis procedure
Comment: Here we face the same problem mentioned in an above comment . I suggest that from herein you replace the terms such “spermatocyte stage” by the days you observed the changes.
Figure 1 legends
Comment: please, correct the scale bar from mm to um.
Lines 114-116: The histochemical examination of the seminiferous tubule of the neogenin- cKO testis after hematoxylin and eosin staining displayed a significant reduction in the testicular cells (Figure 2D)
Comment: Change to: The histochemical examination of the seminiferous tubule of the neogenin- cKO testis after hematoxylin and eosin staining displayed a significant reduction in the AREA OCCUPIED BY testicular GERM cells (Figure 2D)
Lines 154-155: We also tested whether apoptosis is responsible for the decrease in the number of spermatogonia under neogenin down-regulation.
Comment: That is the first time in the text that authors mention decreases of spermatogonia. The authors mentioned before a decrease in primary and secondary spermatocytes, which make the interpretation of the results quite confusing.
Lines 259-260: To about 1.0 g of testis tissues, 1.0 ml of Trizol solution (15596026, Invitrogen, Carlsbad, CA, USA) was added to extract mRNAs according to the manufacturer’s instructions.
Comment: replace mRNA for Total RNA.
Author Response
The manuscript entitled “Down-regulation of neogenin is decline of the proliferation and differentiation of spermatogonia at early phase of spermatogenesis” brings up the idea of a novel role of neogenin on spermatogonia proliferation and differentiation. Overall, the experiments presented were well performed, although some important points should be revised and taken in consideration by the authors.
Major Comments:
- Starting by the title, I suggest the author to revise it, since there is no direct cell proliferation analysis presented on the results
- R) Our data (Figure 3B) revealed that Ki67 expression was significantly decreased in the neogenin-cKO testis.
- Please, inform in the text or in form of supplementary material all the sequences and references (when not design by the authors) for the primers used for RT-PCR, qPCR and CRISPR/Cas9 cKO.
- R) We apologize for omitting the supplemental file. We have provided a new supplemental file with the revision.
- Suggested experiment: I strongly suggest performing qPCRs to sustain the affirmation that “level of neogenin expression varied over the postnatal spermatogenesis stages, from low to very high”. Taking the RT-PCR image as base, I disagree that neogenin expression goes from low to VERY high. A qPCR assay would, quantitatively, answer this point.
- R) We have checked and updated the qPCR data. mRNA expression of neogenin increases because the number of spermatogonia increases during puberty from postnatal day 18 to 24. And change word in the text.
- Lines 81-82: primary and secondary spermatocyte stages reaching the highest level between postnatal days 18 and 24, which corresponds to the spermiogenesis stage.
Comment: If that’s the case, I strongly suggest a new figure containing high magnification (1000X) versions of the immunohistochemistry, now including day 18.
- R) We attempted to update the imaging. However, imaging at 10,000× magnification shows only a few cells and cannot distinguish each phase of spermatogenesis or determine which phase neogenin is expressed in or which phase cells are arrested in upon knock-out of neogenin. A magnification of 10,000× is too high for these purposes. Spermatogenesis has three major phases: spermatogonia, spermatocytes, and spermatids (sperm). Images at 10,000× magnification only show some of these cells.
- Suggested experiment: Lines 90-93: To further verify these conjectures, the localization of neogenin expression in the seminiferous tubules were carefully examined by immunohistochemistry. The results shown in Figure 1D revealed that DAB positive stained on a male germ cell, indicating that neogenin is mainly expressed in spermatogonial stem cells at the early phase of the spermatogenesis.
Comment: The immunohistochemistry does not answer the question if neogenin is implicated in germ cell proliferation. For that I suggest an experiment of cell proliferation (BrDU incorporation, for instance) using your cKO against the wild type.
- R) Thank you for your comment. We have used Ki97 instead of BrdU and updated Figure 3B.
- Lines 118-119: Shown in Figure 2D is that neogenin-cKO dramatically decreased both the primary and secondary spermatocytes in the seminiferous tubules compared with control groups. In addition, the immunohistochemical data shown in Figure 1D and 2D indicated that neogenin specifically localized at the spermatogonia germ cells, and the downregulation of neogenin by neogenin-cKO deterred early meiotic processing of the primary to the secondary spermatocytes phase, strongly implying that neogenin is involved in the early phase of the spermatogenesis differentiation process.
Suggestion: It is EXTREMELY important to show high magnification figures when affirming that a certain type o germ cell is missing, or it is expressing neogenin. Reformulate figures 1E and 2E.
- R) Thank you for your comment. We replacement new images (figure 1E, 2E) in the figure regarding the specific phase of spermatogenesis again. High-magnification figures cannot confirm the localization of neogenin in the edge of the seminiferous tubule.
- Figure 3: The qPCR results for OCT3/4 and SOX2 are not corroborated by the results of Western Blot.
- R) We performed the experiments three times with three different groups and found a similar pattern. Maybe protein and gene expression of Oct3/4 and Sox2 was not corroborated during the spermatogenesis. but it’s no completely different expression pattern between gene and protein of oct3/4 and SOX2 which is similar up and down pattern between gene and protein.
Round 2
Reviewer 1 Report
The revised manuscript shows considerable improvement, but careful attention needs still to be paid to certain points. Without that the manuscript cannot be accepted.
1. Please take into account the neogenin staining seen also in interstitial cells (now clearly seen in new version of the Figure 1).
2. To avoid confusion about number of samples in each experiment, add the number of items on the top of each column (Figs. 1C, 2C, 3A, 4B).
3. Complete the Figure legend 2 (2E is missing).
4. Check all the references carefully. For example, lanes 197-201: “In addition, neogenin, a
5. receptor expressed in type A spermatogonia, is co-expressed with Oct3/4 and thereby regulates the epigenetic state of undifferentiated cells [21, 27]. In summary, expression of neogenin is commonly used together with that of several stem cell factors as a marker of undifferentiated spermatogonia [11,19,23].”
a. Reference 21: neither neogenin nor epigenesis has been mentioned.
b. Reference 27: I had access only to the abstract, but according to it the article shows co-expression of Plzf (an epigenetical regulator) and Oct4. Based on that it cannot be concluded that neogenin would regulates the epigenetic state of testicular cells.
c. References 19 and 23: testis or spermatogenesis has not been discussed in the articles.
d. Reference 11: expression of neogenin has been shown in developing testis, but not as a marker of undifferentiated spermatogonia.
1. End of the discussion: “Therefore, cKO of neogenin in spermatogonia may lead to loss of signaling necessary for maintenance of stem cells, causing spermatogenesis to arrest in germ cells. In support of this, mRNA and protein expression of neogenin in the human testis has been documented in GTEx (https://www.gtexportal.org/ home/gene/NEO1) and a proteomics database (https://www.proteinatlas.org/ENSG00000278646-AC008162.2/tissue/testis#img). Therefore, neogenin may be a useful biomarker to clinically diagnose arrest of spermatogonia.”
a. Mere presence of neogenin also in human testis does not prove the role of neogenin in maintenance of stem cells. The sentence needs rephrasing (if it is necessary at all).
b. Neither of the databases present specific expression of neogenin in spermatogonia. As a matter of fact, the most examples in Protein Atlas show low and weak expression on neogenin in seminiferous ducts in general and when expression is presented, it is not limited to spermatogonia. Instead, more abundant expression is seen in human Leydig cells.
c. How neogenin could be used as a biomarker for spermatogonial arrest? That would need a biopsy and the arrest can be directly seen from the morphology. What kind of extra value neogenin staining would bring?
Minor
1. Explain acronyms (DAB etc.)
Author Response
The revised manuscript shows considerable improvement, but careful attention needs still to be paid to certain points. Without that the manuscript cannot be accepted.
1. Please take into account the neogenin staining seen also in interstitial cells (now clearly seen in new version of the Figure 1).
2. To avoid confusion about number of samples in each experiment, add the number of items on the top of each column (Figs. 1C, 2C, 3A, 4B).
R) 1C / 3A (n=3)
3. Complete the Figure legend 2 (2E is missing).
R) Sorry the missing. We updated in the legend regarding figure 2E
4. Check all the references carefully. For example, lanes 197-201: “In addition, neogenin, a receptor expressed in type A spermatogonia, is co-expressed with Oct3/4 and thereby regulates the epigenetic state of undifferentiated cells [21, 27]. In summary, expression of neogenin is commonly used together with that of several stem cell factors as a marker of undifferentiated spermatogonia [11,19,23].”
a. Reference 21: neither neogenin nor epigenesis has been mentioned.
b. Reference 27: I had access only to the abstract, but according to it the article shows co-expression of Plzf (an epigenetical regulator) and Oct4. Based on that it cannot be concluded that neogenin would regulates the epigenetic state of testicular cells.
c. References 19 and 23: testis or spermatogenesis has not been discussed in the articles.
d. Reference 11: expression of neogenin has been shown in developing testis, but not as a marker of undifferentiated spermatogonia.
R) thank for your great point. We check again and clearly updated all of reference.
1. End of the discussion: “Therefore, cKO of neogenin in spermatogonia may lead to loss of signaling necessary for maintenance of stem cells, causing spermatogenesis to arrest in germ cells. In support of this, mRNA and protein expression of neogenin in the human testis has been documented in GTEx (https://www.gtexportal.org/ home/gene/NEO1) and a proteomics database (https://www.proteinatlas.org/ENSG00000278646-AC008162.2/tissue/testis#img). Therefore, neogenin may be a useful biomarker to clinically diagnose arrest of spermatogonia.”
a. Mere presence of neogenin also in human testis does not prove the role of neogenin in maintenance of stem cells. The sentence needs rephrasing (if it is necessary at all).
R) we replaced new sentence in the text.
b. Neither of the databases present specific expression of neogenin in spermatogonia. As a matter of fact, the most examples in Protein Atlas show low and weak expression on neogenin in seminiferous ducts in general and when expression is presented, it is not limited to spermatogonia. Instead, more abundant expression is seen in human Leydig cells.
R) We tried to the several blocking methods during the immunohistochemistry but always shows non-specific staining on the Leydig cells even negative control.
c. How neogenin could be used as a biomarker for spermatogonial arrest? That would need a biopsy and the arrest can be directly seen from the morphology. What kind of extra value neogenin staining would bring?
R) It is just a hypothesis and potential based on my opinion. And we rewrote that sentence as “Therefore, neogenin needs further study whether involved or not in the arrest of human spermatogonia for overcoming male infertility.”
Minor
1. Explain acronyms (DAB etc.)
R) we put in the manuscript about DAB(3,3'-diaminobenzidine)
Reviewer 2 Report
In table 2, CRISPR/Cas9 is spelled wrongly. It can be corrected during reading proof.
Author Response
In table 2, CRISPR/Cas9 is spelled wrongly. It can be corrected during reading proof.
R) synthesis replaced by synthetic
Round 3
Reviewer 1 Report
Please do the corrections previously required:
1. Reformulate the figures (Figs. 1C, 2C, 3A, 4B) thus that number of values to form each column is shown on top of each column.
2. Rewrite/check with a native English speaker the sentences added to the version 3. In particular, the lines 200-202 are fuzzy.
3. Neogenin staining is detected in Leydig cells by the authors as well as independently in human testis. How can the authors conclude that their result is due to unspecific staining? It is more propriate to mention that staining is also seen in interstitial cells.
Author Response
Reformulate the figures (Figs. 1C, 2C, 3A, 4B) thus that number of values to form each column is shown on top of each column.
R) I understand your comment. Put in the number on top of each column.
Rewrite/check with a native English speaker the sentences added to the version 3. In particular, the lines 200-202 are fuzzy.
R) I re-wrote that sentence. But we have twice the time edited by a professional native speaker. If you want to, I can upload the certificate of native English editing.
Neogenin staining is detected in Leydig cells by the authors as well as independently in human testis. How can the authors conclude that their result is due to unspecific staining? It is more propriate to mention that staining is also seen in interstitial cells.
R) We purchase human-origin synthetic peptide immunized antibodies. It is a species issue for immunoreaction between antibodies and antigens. Negative control slides also reveal DAB signals in the Leydig cells. And we tried three different companies of anti-neogenin antibodies for this study. Unfortunately, only human-origin synthetic peptide immunized antibody is working and optimized for immunohistochemistry in the mice testis. Other antibodies have no signal after immunohistochemistry.